

# Testing the effectiveness of *rbcLa* DNA-barcoding for species discrimination in tropical montane cloud forest vascular plants (Oaxaca, Mexico) using BLAST, genetic distance, and tree-based methods

Sonia Trujillo-Argueta, Rafael F. del Castillo and Abril Velasco-Murguía

CIIDIR Oaxaca, Instituto Politécnico Nacional, Xoxocotlán, Oaxaca, Mexico

Corresponding author
Rafael F. del Castillo,
fsanchez@ipn.mx

## ABSTRACT

DNA-barcoding is a species identification tool that uses a short section of the genome that provides a genetic signature of the species. The main advantage of this novel technique is that it requires a small sample of tissue from the tested organism. In most animal groups, this technique is very effective. However, in plants, the recommended standard markers, such as *rbcL*a, may not always work, and their efficacy remains to be tested in many plant groups, particularly from the Neotropical region. We examined the discriminating power of *rbcL*a in 55 tropical cloud forest vascular plant species from 38 families (Oaxaca, Mexico). We followed the CBOL criteria using BLASTn, genetic distance, and monophyly tree-based analyses (neighbor-joining, NJ, maximum likelihood, ML, and Bayesian inference, BI). *rbcL*a universal primers amplified 69.0% of the samples and yielded 91.3% bi-directional sequences. Sixty-three new *rbcL*a sequences were established. BLAST discriminates 80.8% of the genus but only 15.4% of the species. There was nil minimum interspecific genetic distances in *Quercus, Oreopanax*, and *Daphnopsis*. Contrastingly, Ericaceae (5.6%), Euphorbiaceae (4.6%), and Asteraceae (3.3%) species displayed the highest within-family genetic distances. According to the most recent angiosperm classification, NJ and ML trees successfully resolved (100%) monophyletic species. ML trees showed the highest mean branch support value (87.3%). Only NJ and ML trees could successfully discriminate *Quercus* species belonging to different subsections: *Quercus martinezii* (white oaks) from *Q. callophylla* and *Q. laurina* (red oaks). The ML topology could distinguish species in the Solanaceae clade with similar BLAST matches. Also, the BI topology showed a polytomy in this clade, and the NJ tree displayed low-support values. We do not recommend genetic-distance approaches for species discrimination. Severe shortages of *rbcL*a sequences in public databases of neotropical species hindered effective BLAST comparisons. Instead, ML tree-based analysis displays the highest species discrimination among the tree-based analyses. With the ML topology in selected genera, *rbcL*a helped distinguish infra-generic taxonomic categories, such as subsections, grouping affine species within the same genus, and discriminating species. Since the ML phylogenetic tree could discriminate 48 species out of our 55 studied species, we recommend this approach to resolve tropical montane cloud

forest species using *rbcL*a, as an initial step and improve DNA amplification methods.

# INTRODUCTION

A biodiversity inventory is crucial as a first step to protecting species and ecosystems. A significant portion of global biodiversity remains unnamed. Recent estimations indicate that 8.7 million species of multicellular organisms occur on Earth, but about 20% of those species have been described using morphological approaches since 1750 (*Mora et al., 2011*). Thus, it is urgent to speed up the species identification process (*Hvistendahl, 2016*). In the early 21$^{st}$ century a molecular technique DNA barcoding, was proposed to identify species using short-standardized sequences and only requiring a small sample of tissue (*Hebert et al., 2003*). Cytochrome oxidase 1 (*CO1*) successfully discriminates against many animal species but does not resolve plant species. The Consortium for the Barcode of Life's (CBOL) plant working group evaluated several plastid DNA regions based on universality, sequence quality, and species discrimination, recommending using a core of a two-locus combination of partial genes *rbcL*a + *mat*K as the plant barcode (*CBOL Plant Working Group1 et al., 2009*). Such a universality has not been found in all plant groups, and other studies suggest using additional loci (*Kress & Erickson, 2007*; *Fazekas et al., 2008*; *China Plant BOL Group et al., 2011*; *Pang et al., 2012*). Moreover, *mat*K may work very well for orchid species (*Lahaye et al., 2008*) but not for certain fern groups (*Trujillo-Argueta et al., 2021*). Furthermore, in some angiosperm genera, such as *Salix* (*Percy et al., 2014*) and *Quercus* (*Piredda et al., 2011*), plastid markers might not work at all.

On average, the resolution of the tested DNA barcoding markers for plants is not as high as barcode markers used for many animal groups (*Fazekas et al., 2008*; *CBOL Plant Working Group, 2009*). Of the possible plant markers, *rbcL*a appears to be one of the best plant barcodes, because of its successful amplification and sequencing. Although far from perfect, the resolution of *rbcL*a was shown to be better than *mat*K when tested both barcodes in wild arid plants in the United Arab Emirates (*Maloukh et al., 2017*) and when tested alone, in plants of Saudi Arabia (*Bafeel et al., 2012*). Also, *rbcL*a can be a valuable tool to identify species in conditions in which other methods are impractical. For instance, this marker was successfully used for studying root diversity patterns in old-field communities in Ontario, Canada (*Kesanakurti et al., 2011*). This kind of research is encouraging, but more studies are needed to explore the resolution potential of this marker for species in ecosystems other than those of temperate regions. The Neotropics are considered the richest region in biodiversity (*Gaston & Williams, 1996*; *Thomas, 1999*). Several barcoding studies have been performed in neotropical animals (*e.g.*, *Hajibabaei et al., 2006*). However, barcoding plant studies in this area are scarce. The available studies

are often limited to a few plant groups such as orchids (*Lahaye et al., 2008*) or ferns (*Nitta, 2008*; *Trujillo-Argueta et al., 2021*).

In Mexico, the tropical montane cloud forest (TMCFs) is a top priority ecosystem for conservation due to its high diversity, endemism richness, and anthropogenic threats (*Villaseñor, 2010*; *Toledo-Aceves et al., 2011*). Due to the reproductive biology of plants, the universality of DNA barcodes has been difficult to achieve when dealing with several taxa, therefore, some authors suggest to develop a DNA barcode library locally to be used for conservation and ecological studies (*Lahaye et al., 2008*; *De Groot et al., 2011*). Our study is part of a long-term project to characterize DNA barcodes of tropical plants of southern Mexico. We choose *rbcL*a as the first marker to study because (1) it is widely used in phylogenic analyses, (2) it has been helpful for ecological studies (*i.e.*, *Kesanakurti et al., 2011*), and is one of the markers with more published sequences in public databases. The aim of this study is to evaluate the performance of the plant core DNA barcode *rbcL*a, using universal primers for vascular plants, as a first stage of DNA barcoding analysis in an unexplored tropical montane cloud forest of the Mixteca Baja, Oaxaca, Mexico. We followed the three above-mentioned CBOL criteria and built a barcode library of native plant species for this region. Sequences obtained of these species were submitted to The Barcode of Life Data System (BOLD) and GenBank. BOLD is a bioinformatic workbench devoted to acquiring, storing, analyzing, and publishing DNA barcode records (*Ratnasingham & Hebert, 2007*).

# MATERIALS AND METHODS

## Description of study site

We conducted the field study in a tropical montane cloud forest at different successional stages in the San Miguel Cuevas, Santiago Juxtlahuaca Municipality, Mixteca Baja (17°15′00.96″N, 98°02′57.34″, centroid coordinates), which belongs to the Western physiography region of the state of Oaxaca, in southern Mexico (*Ortíz-Pérez, Hernández Santana & Figueroa Mah-Eng, 2004*). The climate is semi-humid, temperate to semi-warm (1,382 mm and 16.8 °C, mean annual precipitation and temperature, respectively, *Fernandez-Eguiarte, Zavala-Hidalgo & Romero-Centeno, 2020*), with soils rich in organic matter, steep topography, and a mean altitude of 2,187 m.

## Studied species

We carried out random walks thought the forest area and collected one hundred samples of fresh vascular plants, which position was georeferenced with a GPS. In the field, we took digital photographs of each sampled plant and its main structures. The number of samples collected per taxon was one and occasionally two. A small sample of fresh leaf tissue (2–5 g) was also collected from the same specimen, placed in a sealed plastic bag, and kept fresh until stored at −20 °C in a lab freezer. Voucher plants were pressed flat for standard herborization (drying, sanitizing, identification, mounting, labelling, and shelf-storing). The voucher specimens were deposited at the herbarium of CIIDIR Oaxaca, Instituto Politécnico Nacional (OAX), pending for registration numbers due to the pandemic crisis. The species analyzed, and their IUCN Red List Status is shown on Table 1. The municipal

**Table 1 Species of the tropical montane cloud forest of San Miguel Cuevas, Santiago Juxtlahuaca Municipality, Oaxaca, Mexico, used in this study.**

| Sample ID | Family | Morphological species | IUCN status | bp lenght | BOLD Process ID | GeneBank Accession No. |
|---|---|---|---|---|---|---|
| SMC7 | Rubiaceae | *Deppea grandiflora* Schltdl. | LC | 553 | TFOAX001-19 | ON002500 |
| SMC10 | Styracaceae | *Styrax glabrescens* Benth. | DD | 535 | TFOAX002-19 | ON002540 |
| SMC19 | Thymelaeaceae | *Daphnopsis selerorum* Gilg | LC | 535 | TFOAX003-19 | ON002498 |
| SMC28 | Solanaceae | *Solanum nigricans* M. Martens & Galeotti | LC | 553 | TFOAX004-19 | ON002539 |
| SMC29 | Polygalaceae | *Monnina xalapensis* Kunth | LC | 553 | TFOAX005-19 | ON002512 |
| SMC37 | Ericaceae | *Comarostaphylis longifolia* (Benth.) Klotzsch | DD | 540 | TFOAX006-19 | ON002495 |
| SMC41 | Rubiaceae | *Deppea grandiflora* Schltdl. | LC | 553 | TFOAX007-19 | ON002501 |
| SMC61 | Euphorbiaceae | *Tragia* aff. *nepetifolia* Cav. | DD | 541 | TFOAX008-19 | ON002541 |
| SMC70 | Rubiaceae | *Hoffmannia longipetiolata* Pol. | DD | 553 | TFOAX009-19 | ON002505 |
| SMC75 | Olaceae | *Osmanthus americanus* (L.) Benth. & Hook. f. ex A. Gray | DD | 553 | TFOAX010-19 | ON002521 |
| SMC76 | Commelinaceae | *Commelina coelestis* Willd. | DD | 553 | TFOAX011-19 | ON002497 |
| SMC94 | Thymelaeaceae | *Daphnopsis tuerckheimiana* Donn. Sm. | NT | 540 | TFOAX012-19 | ON002499 |
| SMC97 | Rubiaceae | *Deppea guerrerensis* Dwyer & Lorence | DD | 536 | TFOAX013-19 | ON002502 |
| SMC99 | Solanaceae | *Physalis philadelphica* Lam. | LC | 535 | TFOAX014-19 | ON002525 |
| SMC104 | Apocinaceae | *Vallesia aurantiaca* (M. Martens & Galeotti) J.F. Morales | DD | 553 | TFOAX015-19 | ON002545 |
| AVM2 | Betulaceae | *Alnus acuminata* Kunth | LC | 560 | DVHTF001-19 | ON002486 |
| AVM12 | Berberidaceae | *Berberis lanceolata* Benth. | DD | 557 | DVHTF002-19 | ON002489 |
| AVM13 | Araliaceae | *Oreopanax sanderianus* Hemsl. | VU | 557 | DVHTF003-19 | ON002519 |
| AVM14 | Scrophulariaceae | *Buddleja cordata* Kunth | LC | 557 | DVHTF004-19 | ON002490 |
| AVM15 | Convolvulaceae | *Ipomoea elongata* Choisy | DD | 557 | DVHTF005-19 | ON002506 |
| AVM16 | Solanaceae | *Solanum hispidum* Pers | DD | 553 | DVHTF006-19 | ON002537 |
| AVM17 | Solanaceae | *Solandra maxima* (Sessé & Moc.) P.S. Green | DD | 558 | DVHTF007-19 | ON002536 |
| AVM27 | Commelinaceae | *Commelina coelestis* Willd. | DD | 557 | DVHTF008-19 | ON002496 |
| AVM30 | Lycopodeaceae | *Lycopodium clavatum* L. | LC | 560 | DVHTF009-19 | MZ771330 |
| AVM32 | Primulaceae | *Myrsine juergensenii* (Mez) Ricketson & Pipoly | LC | 557 | DVHTF010-19 | ON002516 |
| AVM33 | Ericaceae | *Vaccinium leucanthum* Schltdl. | DD | 558 | DVHTF011-19 | ON002544 |
| AVM34 | Fagaceae | *Quercus martinezii* C.H. Mull. | LC | 560 | DVHTF012-19 | ON002531 |
| AVM35 | Fagaceae | *Quercus laurina* Bonpl. | LC | 558 | DVHTF013-19 | ON002530 |
| AVM36 | Melastomataceae | *Miconia glaberrima* (Schltdl.) Naudin | LC | 545 | DVHTF014-19 | ON002511 |
| AVM40 | Pteridaceae | *Adiantum andicola* Liebm. | DD | 557 | DVHTF015-19 | ON002485 |
| AVM43 | Rosaceae | *Rubus sapidus* Schltdl. | DD | 557 | DVHTF016-19 | ON002534 |
| AVM45 | Araliaceae | *Oreopanax xalapensis* (Kunth) Decne. & Planch. | LC | 556 | DVHTF017-19 | ON002520 |
| AVM46 | Piperaceae | *Piper umbellatum* L. | DD | 557 | DVHTF018-19 | ON002528 |
| AVM47 | Lauraceae | *Ocotea helicterifolia* (Meisn.) Hemsl. | DD | 557 | DVHTF019-19 | ON002517 |
| AVM48 | Fabaceae | *Calliandra houstoniana* (Mill.) Standl. | LC | 558 | DVHTF020-19 | ON002491 |
| AVM50 | Lauraceae | *Ocotea helicterifolia* (Meisn.) Hemsl. | DD | 553 | DVHTF021-20 | ON002518 |
| AVM52 | Primulaceae | *Parathesis donnell-smithii* Mez | LC | 553 | DVHTF022-20 | ON002522 |
| AVM53 | Fagaceae | *Quercus calophylla* Schltdl. & Cham. | LC | 553 | DVHTF023-20 | ON002529 |
| AVM54 | Euphorbiaceae | *Cnidoscolus aconitifolius* (Mill.) I.M. Johnst. | LC | 553 | DVHTF024-20 | ON002494 |
| AVM59 | Amaranthaceae | *Iresine diffusa* Humb. & Bonpl. ex Willd. | DD | 553 | DVHTF025-20 | ON002507 |

| Sample ID | Family | Morphological species | IUCN status | bp lenght | BOLD Process ID | GeneBank Accession No. |
|---|---|---|---|---|---|---|
| AVM62 | Solanaceae | *Solanum nigricans* M. Martens & Galeotti | LC | 553 | DVHTF026-20 | ON002538 |
| AVM64 | Rubiaceae | *Arachnothryx buddleioides* (Benth.) Planch. | LC | 553 | DVHTF027-20 | ON002488 |
| AVM65 | Passifloraceae | *Passiflora quadraticordata* Lozada-Pérez | DD | 553 | DVHTF028-20 | ON002524 |
| AVM67 | Urticaceae | *Urera killipiana* Standl. & Steyerm. | LC | 553 | DVHTF029-20 | ON002543 |
| AVM69 | Rubiaceae | *Hoffmannia longipetiolata* Pol. | DD | 553 | DVHTF030-20 | ON002504 |
| AVM71 | Gesneriaceae | *Moussonia deppeana* (Schltdl. & Cham.) Hanst. | DD | 553 | DVHTF031-20 | ON002515 |
| AVM73 | Pinaceae | *Pinus montezumae* Lamb. | LC | 553 | DVHTF032-20 | ON002526 |
| AVM78 | Dicksoniaceae | *Lophosoria quadripinnata* (J.F. Gmel.) C. Chr. | DD | 553 | DVHTF033-20 | ON002509 |
| AVM79 | Passifloraceae | *Passiflora quadraticordata* Lozada-Pérez | DD | 553 | DVHTF034-20 | ON002523 |
| AVM80 | Cyperaceae | *Rhynchospora aristata* Boeck. | DD | 553 | DVHTF035-20 | ON002532 |
| AVM82 | Marattiaceae | *Marattia weinmanniifolia* Liebm. | DD | 553 | DVHTF036-20 | ON002510 |
| AVM83 | Cupressaceae | *Juniperus flaccida* Schltdl. | LC | 553 | DVHTF037-20 | ON002508 |
| AVM84 | Pinaceae | *Pinus pseudostrobus* Lindl. | LC | 553 | DVHTF038-20 | ON002527 |
| AVM87 | Asteraceae | *Montanoa tomentosa* Cerv. | DD | 553 | DVHTF039-20 | ON002513 |
| AVM89 | Meliaceae | *Guarea glabra* Vahl | LC | 553 | DVHTF040-20 | ON002503 |
| AVM90 | Meliaceae | *Trichilia havanensis* Jacq. | LC | 553 | DVHTF041-20 | ON002542 |
| AVM93 | Rubiaceae | *Arachnothryx buddleioides* (Benth.) Planch. | LC | 553 | DVHTF042-20 | ON002487 |
| AVM95 | Gesneriaceae | *Moussonia deppeana* (Schltdl. & Cham.) Hanst. | DD | 553 | DVHTF043-20 | ON002514 |
| AVM98 | Solanaceae | *Cestrum commune* C.V. Morton ex Mont.-Castro | DD | 553 | DVHTF044-20 | ON002492 |
| AVM100 | Poaceae | *Zeugites hintonii* Hartley | DD | 553 | DVHTF045-20 | ON002546 |
| AVM101 | Asteraceae | *Roldana angulifolia* (DC.) H.Rob. & Brettell | DD | 553 | DVHTF046-20 | ON002533 |
| AVM103 | Verbenaceae | *Citharexylum hexangulare* Greenm. | DD | 474 | DVHTF047-20 | ON002493 |
| AVM106 | Lamiaceae | *Salvia clarkcowanii* B.L. Turner | DD | 553 | DVHTF048-20 | ON002535 |

Notes:
*The IUCN Red List (2021)* status, DNA length obtained with *rbcLa* barcode, BOLD Process ID, and GenBank Accession numbers are also shown.
LC, Least Concern; DD, Data Deficit; NT, Near Threatened; VU, Vulnerable.

councils of San Miguel Cuevas granted permission to conduct our field studies on their lands. Two commissioners of the communal property, Mr. Pedro Gil (2017–July 2018) and Mr. Damián Domínguez (July 2018 to June 2019) were directly responsible for such permissions. During the field trips, Mr. Heladio Luna Rodríguez, a member of The San Miguel Cuevas Community local authority, supervised, guided, and helped us throughout the sampling process. In no case was the entire plant collected; collecting the samples did not kill the plants, which were left alive in their original places. Based on The International Union for Conservation of Nature (IUCN) Red List (accessed December 19th, 2021) from all 55 species in this study, more than half 52.73% (29/55) were not previously registered (Data Deficit DD), 43.64% (24/55) belong to the Least Concern (LC) category and 3.64% (2/55) hold some type of concern; *Daphnopsis tuerckheimiana* the status of Near Threatened (NT) and *Oreopanax sanderianus* of Vulnerable species (VU).

Fresh plant vouchers were first identified to family level and then determined by the following specialists to species level: Daniel Tejero-Díez, UNAM FES Iztacala, México, lycopod and ferns; Sergio Zamudio, Institute of Ecology, Veracruz, México, Berberidaceae;

Rafael F. del Castillo, IPN CIIDIR Oaxaca, México, Pinaceae; Jesús Guadalupe González Gallegos, University of Guadalajara, México, Lamiaceae; Socorro González Elizondo, IPN CIIDIR Durango, México, Cyperaceae and Ericaceae; Susana Valencia Avalos, UNAM Facultad de Ciencias, México, Fagaceae; J.R. Kuethe, University of Auckland, New Zealand, Passifloraceae; and Rufina García, IPN CIIDIR Oaxaca, México, the rest of the families.

## DNA amplification and sequencing

Genomic DNA was extracted from 2 mg leaf tissue with FastDNA SPIN kit and FastPrep® (MP Biomedicals, Santa Ana, CA, USA) equipment. DNA concentration (ng/µl) and purity (260/280 A) from the genomic DNA extracted were measured with a Biophotometer (Eppendorf®). Plant core barcoding partial gene *rbcL*a (670 bp) was used for amplification. We used standard primers from the Canadian Center for DNA Barcoding (CCDB) (*Kuzmina, 2011*), *rbcL*a-F ATGTCACCACAAACAGAGACTAAAGC (*Tate & Simpson, 2003*) and *rbcL*a-R GTAAAATCAAGTCCACCRCG (*Kress & Erickson, 2007*). *rbcL*a was amplified using a 25 µL volume of reaction mixture: 15.8 µL of nuclease-free water, 5 µl MyTaq Buffer reaction (kit MyTaqDNA Polymerase Bioline), 1 µL of forward primer, 1 µL of reverse primer, 0.2 µL of MyTaq Polymerase and 2 µL of isolated genomic DNA template (39± ng/µL mean concentration ±5.13 EE). PCR reaction was carried out using an Applied Biosystems Veriti® thermocycler. We followed *Fazekas et al. (2012)* protocols for *rbcL*a amplification. The PCR temperature cycling program was: 94 °C for 4 min; 35 cycles of 94 °C for 30 s, 55 °C for 30 s, 72 °C for 1 min; final extension of 72 °C during 10 min. *rbcL*a gene length is 1,428 bp, but barcode method uses only 670 bp of this gene, known as *rbcL*a (*Dong et al., 2014*). Amplified PCR products were detected using agarose gel electrophoresis (1.2% agarose gel TBE) under UV light by staining with GelRed Nucleic Acid (Biotium, Fremont, CA, USA). PCR products were purified using the EZ-10 Spin Column PCR Products Purification Kit (Biobasic, Markham, Canada). All PCR products were sequenced by Capillary Electrophoresis Sequencing (CES) in an ABI 3130xl Genetic Analyser at the Laboratorio Bioquímica Molecular UBIPRO FES Iztacala UNAM, or with an AB3730 at the Laboratorio de Servicios Genómicos LANGEBIO-CINVESTAV, Mexico.

## DNA alignment

*rbcL*a sequence chromatograms were assembled into contigs using CodonCode Aligner v.9.0.1 http://www.codoncode.com/aligner, and the resulting nucleotide sequences were manually edited. Consensus sequences were generated and aligned using MUSCLE (*Edgar, 2004*). These alignments were examined by eye and corrected in case of base ambiguity.

## BOLD and genebank

Three files were included in the metadata submitted to BOLD: (1) Specimen data file including detailed voucher information, scientific names of the taxa sampled, collection dates, geographical coordinates, elevation, collectors, identifiers, and habitat. (2) An image file was submitted with high-quality specimen images from each plant. (3) A trace file was

submitted along with primers and the direction of sequences. DNA sequences, edited, aligned, using CodonCode software, and in FASTA format, were uploaded to BOLD and referenced by Sample IDs. Metadata and DNA sequences submitted to BOLD were registered under project name "Diversity of a humid temperate forest in Oaxaca, Mexico" project code DVHTF (http://www.boldsystems.org). DNA sequences were also submitted to the GenBank.

## Species differentiation

We used three strategies to evaluate species discrimination:

(a) The Basic Local Alignment Search Tool for nucleotide (BLASTn) method (*Altschul et al., 1990*). This tool compares the query sequence against the GenBank sequence database available online by the National Center for Biotechnology Information (NCBI) https://www.ncbi.nlm.nih.gov. Identification at the genus level was considered successful when all hits with the maximum percent identity scores >99% involved a single genus. Species identification was considered successful only when the highest maximal percent identity included a single species and scored >99% (*Bafeel et al., 2012*; *Abdullah, 2017*).

(b) Genetic divergence. Interspecific and intraspecific distances were obtained in MEGAX (*Kumar et al., 2018*). Genetic distance was inferred from 1,000 replicates, and the evolutionary distances were computed using the Kimura two-parameter method with gaps/missing data treatment adjusted using pairwise deletion. The K2P genetic distances percentages of families, genera, and species were analyzed in the Barcode of Life Data Systems (BOLD, www.boldsystems.org) (*Ratnasingham & Hebert, 2007*).

(c) Monophyly tree-based analyses using Neighbor-Joining (NJ), Maximum Likelihood (ML), and Bayesian Inference (BI) analysis.

NJ trees were constructed using MEGAX (*Kumar et al., 2018*) inferred from 1,000 replicates, and the evolutionary distances were computed using the Kimura two-parameter method with gaps/missing data treatment adjusted using pairwise deletion. Following *Braukmann et al. (2017)*, we used all species but duplicates to avoid bias created by an unequal number of sequences per species. ML analyses were run on the IQ-TREE web server (http://iqtree.cibiv.univie.ac.at) with settings of automatic substitution model and ultrafast bootstrap analysis. Internal node support was calculated using 1,000 bootstrap replicates. Tree inference using Bayesian analysis was run on MrBayes 3.2.2 on XSEDE *via* the CIPRES supercomputer cluster (www.phylo.org) with two runs, MrBayes block of four chains, 2 h maximum time to run, and the nucleic acid selection settings for this web server. The resultant ML and BI trees were visualized in the interactive Tree of Life (iTOL) (*Letunic & Bork, 2019*). We evaluated which of the tree-based methods (NJ, ML, and MB) recovered more monophyletic species with a bootstrap/posterior probabilities support of >70% (*De Groot et al., 2011*).

## RESULTS

### Studied taxa with DNA amplification and sequencing success

We could successfully amplify 69% (69/100) of the botanical samples collected and obtain high-quality bidirectional sequences (>250 bp) in 91.3% (63/69) of the samples collected,

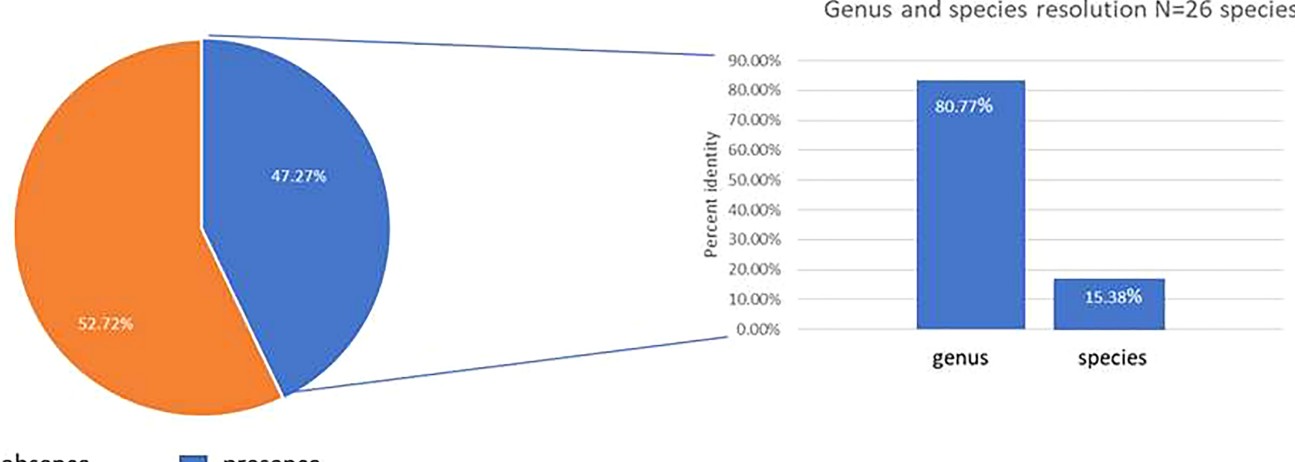

**Figure 1 Graphic BLAST results.** Fraction of the new *rbcL* sequences published in the GenBank (left), and resolution at genus and species levels for those species with previous available sequences released in the GenBank (accessed in March 2022) using BLASTn (right).

using the standard primers of the CCDB for *rbcL*a. The total number of specimens with high-quality bidirectional sequences includes 55 species, of which 29.1% (16/55) were herbs, and 70.9% (39/55) were woody plants (trees and shrubs). These species belong to 38 families and 48 genera. Twenty-seven families include one species, and 11 families include two to five species (Table 1).

## BLAST

Only 47.3% (26/55) of our studied species had a previous *rbcL*a sequence register in the GenBank database (Fig. 1). We contributed in this study with 13 new species in the GenBank Taxonomy Database (*Daphnopsis selerorum, Solanum nigricans, Comarostaphylis longifolia, Daphnopsis tuerckheimiana, Deppea guerrerensis, Vallesia aurantiaca, Myrsine juergensenii, Quercus martinezii, Miconia glaberrima, Passiflora quadraticordata, Cestrum commune, Zeugites hintonii* and *Salvia clarckcowanii*), since none of this species had a previous register with *rbcL*a nor with any other gene sequence. Using previously published records in the GenBank of *rbcL*a sequence for our species (26) and BLASTn, we got 100% resolution in all the families, 80.77% (21/26) in the genera, and only 15.38% (4/26) at the species level (Fig. 1). Just four species, *Monnina xalapensis, Cnidoscolus aconitifolius, Iresine diffusa*, and *Lophosoria quadripinnata*, displayed the highest score BLAST match to a single species with more than 99% identity. Most of our *rbcL*a sequences matched from 2–12 species with ≥99% maximal percent identity; and seven species, *Alnus acuminata, Solanum hispidum, Quercus laurina, Quercus callophylla, Pinus montezumae, Osmanthus americanus*, and *Physalis phyladelphica*, matched the *rbcL*a sequences in the GenBank with >30 different species. The highest score BLAST match for our species are shown in Table 2.

**Table 2 BLAST results.**

| Sample ID | Morphological species | Identity (%) | Best BLAST match | Accession no. |
|---|---|---|---|---|
| AVM2 | *Alnus acuminata* | 100 | Δ *Alnus nepalensis* | NC_039991.1 |
| SMC7 | *Deppea grandiflora* | 99.46 | ○ *Cosmibuena grandiflora* | AM117220.1 |
| AVM12 | *Berberis lanceolata* | 99.82 | ○ *Berberis thunbergii* | KX162895.1 |
| AVM16 | *Solanum hispidum* | 100 | ○ *Solanum torvm* | MN218087.1 |
| SMC29 | *Monnina xalapensis* | **100** | ***Monnina xalapensis*** | AM234184.1 |
| AVM30 | *Lycopodium clavatum* | 100 | ○ *Lycopodium clavatum* | KF977478.1 |
| AVM35 | *Quercus laurina* | 100 | Δ *Quercus phillyraeoides* | NC_048488.1 |
| AVM40 | *Adiantum andicola* | 99.82 | ○ *Adiantum feei* | MH019567.1 |
| AVM46 | *Piper umbellatum* | 100 | ○ *Piper umbellatum* | KF496838.1 |
| AVM52 | *Parathesis donnell-smithii* | 100 | ○ *Stylogyne longifolia* | MF786262.1 |
| AVM53 | *Quercus callophylla* | 100 | Δ *Quercus phillyraeoides* | NC_048488.1 |
| AVM54 | *Cnidoscolus aconitifolius* | **100** | ***Cnidoscolus aconitifolius*** | MZ045411.1 |
| AVM59 | *Iresine diffusa* | **100** | ***Iresine diffusa*** | JQ590112.1 |
| AVM64 | *Arachnothryx buddleioides* | 100 | ○ *Arachnothryx monteverdensis* | JQ594656.1 |
| AVM70 | *Hoffmannia longipetiolata* | 99.64 | ○ *Omiltemia filisepala* | AM117251.1 |
| AVM73 | *Pinus montezumae* | 100 | Δ *Pinus arizonica* | KC156714.1 |
| SMC75 | *Osmanthus americanus* | 99.81 | Δ *Osmanthus americanus* | NC_048503.1 |
| AVM78 | *Lophosoria quadripinnata* | **99.64** | ***Lophosoria quadripinnata*** | MW138175.1 |
| AVM80 | *Rhynchospora aristata* | 99.82 | ○ *Rhynchospora sp* | JQ594519.1 |
| AVM82 | *Marattia weinmanniifolia* | 100 | ○ *Marattia douglasii* | MT657852.1 |
| AVM83 | *Juniperus flaccida* | 100 | ○ *Juniperus flaccida* | HM024304.1 |
| AVM84 | *Pinus pseudostrobus* | 100 | ○ *Pinus flexilis* | MG215114.1 |
| AVM87 | *Montanoa tomentosa* | 100 | ○ *Montanoa tomentosa* | MT189234.1 |
| AVM89 | *Guarea glabra* | 100 | ○ *Ruagea pubescens* | MN454793.1 |
| AVM90 | *Trichilia havanensis* | 99.82 | Meliaceae* *sp.* | EU042974.1 |
| SMC99 | *Physalis philadelphica* | 100 | Δ *Physalis minima* | NC_048515.1 |

**Notes:**
Best BLASTn match found on queries against *rbcL*a nucleotides sequences in the database of GenBank for those species with previously published sequences in the GenBank. E- value, in all cases 0.0.
In bold morphological species corresponding with only the studied species in GenBank database.
○ >2–<30 species with the same highest percent of identity.
Δ >30 species with the same highest identity percent.
* *T. havanensis* showed the best match with another unidentified species of the Meliaceae in the GenBank published sequence.

A specimen data file, image file, and trace file(s) were submitted to BOLD along with edited and aligned sequences for each of our 63 botanical samples (55 species and eight different duplicates) and can be accessed through the BOLD DNA database (http://www.boldsystems.org) under the 'DVHTF' project. Sixty-three new sequences generated by this study for *rbcL*a with their BOLD Process ID, and GenBank Accession numbers, are shown on Table 1.

## Genetic divergence

The distribution of intra- and interspecific K2P distances across all taxon pairs of our 55 species of plants of The Mixteca Baja, Oaxaca, tropical montane cloud forest, obtained

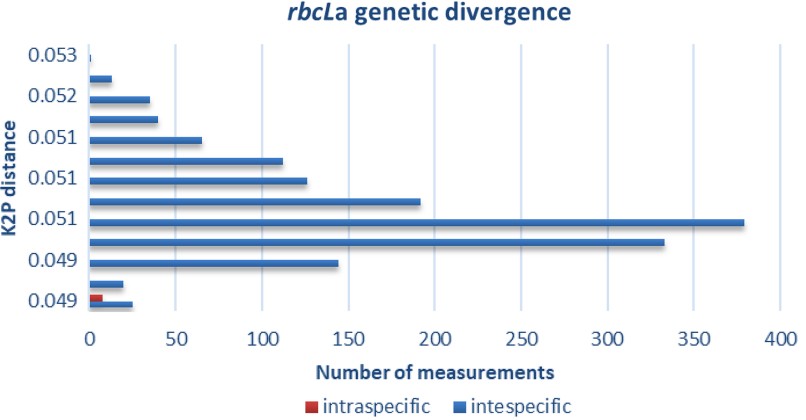

**Figure 2  Genetic divergence.** Intra- and interspecific genetic distances obtained with barcode *rbcL*a in 55 plant species of tropical montane cloud forest in the Mixteca Alta, Oaxaca, Mexico.

**Table 3  Intergeneric *rbcL*a genetic distances of the tropical montane cloud forest species.**

| Genus | Species comparision | | Genetic divergence |
|---|---|---|---|
| *Daphnopsis* | *Daphnopsis tuerckheimiana* TFOAX012-19 | *Daphnopsis selerorum* TFOAX003-19 | 0.0000 |
| *Deppea* | *Deppea guerrerensis* TFOAX013-19 | *Deppea grandiflora* TFOAX001-19 | 1.5076 |
| *Deppea* | *Deppea guerrerensis* TFOAX013-19 | *Deppea grandiflora* TFOAX007-19 | 1.5076 |
| *Oreopanax* | *Oreopanax xalapensis* DVHTF017-19 | *Oreopanax sanderianus* DVHTF003-19 | 0.0000 |
| *Pinus* | *Pinus pseudostrobus* DVHTF038-20 | *Pinus montezumae* DVHTF032-20 | 2.0189 |
| *Quercus* | *Quercus laurina* DVHTF013-19 | *Quercus martinezii* DVHTF012-19 | 0.0000 |
| *Quercus* | *Quercus callophylla* DVHTF023-20 | *Quercus martinezii* DVHTF012-19 | 0.0000 |
| *Quercus* | *Quercus callophylla* DVHTF023-20 | *Quercus laurina* DVHTF013-19 | 0.0000 |
| *Solanum* | *Solanum hispidum* DVHTF006-19 | *Solanum nigricans* TFOAX004-19 | 0.7269 |
| *Solanum* | *Solanum nigricans* DVHTF026-20 | *Solanum hispidum* DVHTF006-19 | 0.7269 |

**Note:**
Intergeneric *rbcL*a genetic distances of the tropical montane cloud forest species in the Mixteca Baja, Oaxaca, Mexico, found in multi-species genera. The Bold Process ID is below the scientific names.

from partial gen *rbcL*a are shown in Fig. 2. Mean pairwise genetic distance within species was 0 (*i.e.*, identical for the tested sequences), within genus 0.65 + 0.07, and 1.76 + 0.03 within families. Congeneric species of *Quercus*, *Daphnopsis*, and *Oreopanax* did not show genetic divergence. Contrastingly, *Solanum*, *Deppea*, and *Pinus* displayed intergeneric differences (Table 3).

**Table 4 Mean *rbcL*a intergeneric and interspecific genetic distances.**

| Family | No. Genera | No. Species | Mean divergence (%) |
|---|---|---|---|
| Asteraceae | 2 | 2 | 3.3344 |
| Ericaceae | 2 | 2 | 5.5723 |
| Euphorbiaceae | 2 | 2 | 4.5869 |
| Meliaceae | 2 | 2 | 0.7286 |
| Primulaceae | 2 | 2 | 1.2784 |
| Rubiaceae | 3 | 4 | 1.6207 |
| Solanaceae | 4 | 5 | 1.6558 |

Note:
Mean *rbcL*a intergeneric and interspecific genetic distances in the multigenera and multispecies families of this study.

The mean genetic divergence observed in the studied families with two or more genera is shown in Table 4. The highest mean divergence values were observed in the Ericaceae, Euphorbiaceae, and Asteraceae families.

## Monophyly tree-based analyses

Phylogenetic tree-based analysis using Neighbor-Joining (Fig. S1), Maximum Likelihood (Fig. 3), and Bayesian Inference tree (Fig. S2) were reconstructed to evaluate our 55 species discrimination using the *rbcL*a barcode region. In all cases, we used ferns and a lycopodium as outgroups. These tree-based methods evaluated which tree rendered the greatest species resolution and whether the barcode sequences generated monophyletic species (Table 5). NJ and ML phylogenetic trees resolved 100% of monophyletic species using *rbcL*a. Nevertheless, the clade support value >70% with a bootstrap of 1,000 replicates yielded the most robust phylogeny in the ML tree (87.3%, 48/55) than the one obtained in the NJ tree (70.9%, 39/55). Therefore, we present the ML phylogenetic tree (Fig. 3). Although the BI tree showed the highest clade support value (92.7%, 51/55), this tree did not resolve all 55 species as monophyletic species; one polytomy was observed in the *Quercus* clade and another in the Solanaceae clade (Fig. S2).

## DISCUSSION

Our study reveals the advantages and limitations of the *rbcL*a barcode region for species identification of vascular plant species of a neotropical montane cloud forest. First, the amplification was not universal, since near 30% of our samples did not amplify. However, bi-directional sequencing was highly successful from those samples that we could amplify. Using BLAST as an identification tool for genus level is convenient as it proved accurate in most cases. However, this was not usually the case for many of the studied species. Finally, in selected genera, this marker helped distinguish infra-generic taxonomic categories, such as subsections, and helps to group affine species within the same genus. Below, we discuss in detail these issues.

Multiple factors can cause the absence of DNA amplification in some samples. Since we could amplify *rbcL*a in several species, the possibilities of methodological failures or problems with the reactants or the lab equipment used are unlikely. One possible cause of

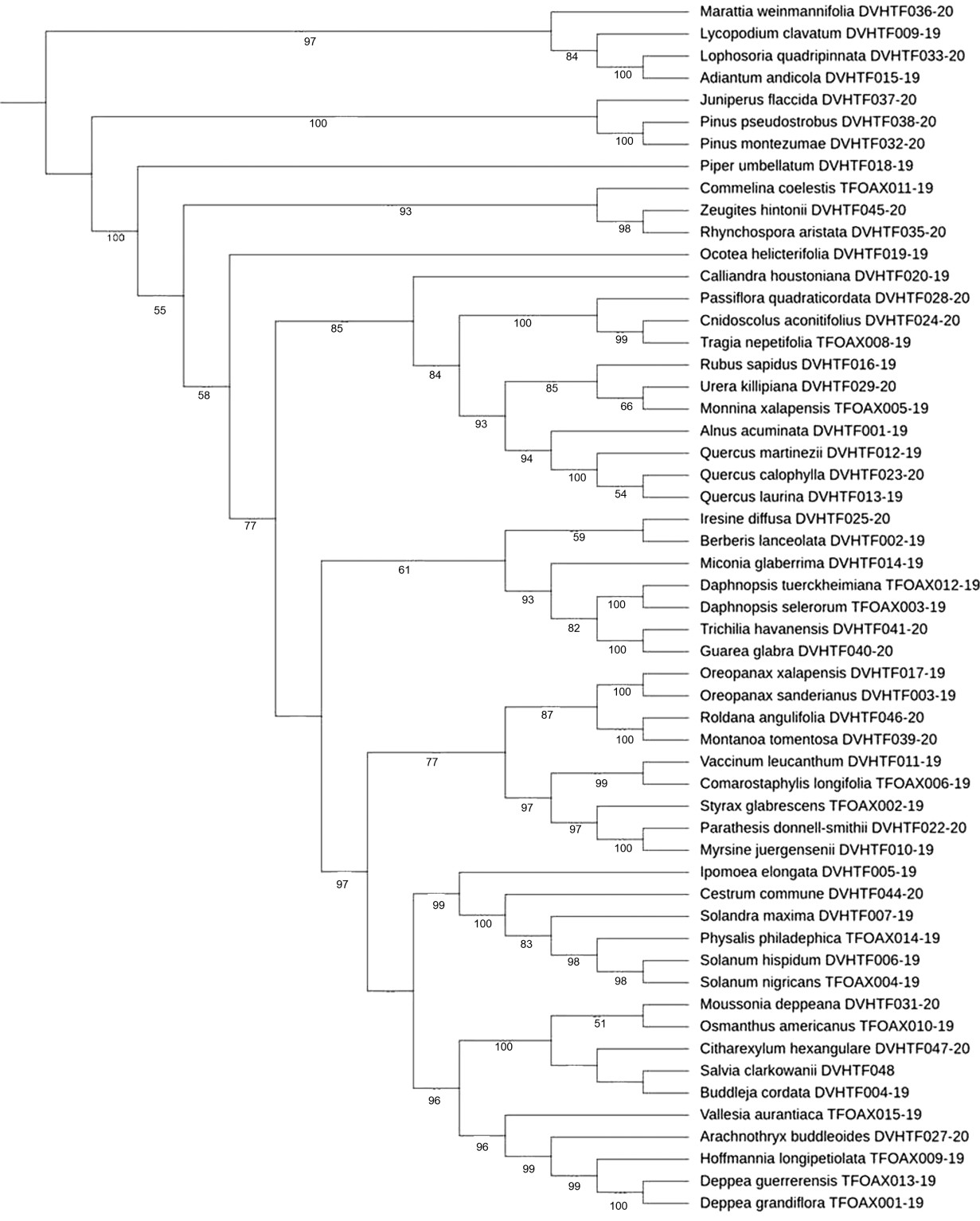

**Figure 3 ML cladogram.** Maximum likelihood cladogram of plant core barcoding gene *rbcLa* for 55 sequences of plants from of Mixteca Baja, Oaxaca, México, tropical montane cloud forest. Bootstrap values based on 1,000 replications are listed as percentages at branching points.

**Table 5 Proportion of resolved monophyletic species and support value obtained (bootstrap/posterior probabilities) with different phylogenetic techniques.**

|  | Neighbor joining tree | Maximum likelihood tree | Bayesian inference tree |
|---|---|---|---|
| Percent of monophyletic species resolved | 100.00 | 100.00 | 85.45 |
| Percent species resolved with support value >70% | 70.91 | 87.27 | 92.73 |

Note:
Proportion of resolved monophyletic species and support value obtained (bootstrap/posterior probabilities) with different phylogenetic techniques using plant core barcoding gene *rbcL*a in 55 studied species of the tropical montane cloud forest, Mixteca Baja, Mexico.

the amplification failure is DNA degradation in some samples, as those were collected in the field and brought to the lab. During this time, the tissues may become degraded in some species. This appears to be a plausible explanation for cases in which DNA from tissue samples was successfully amplified in one individual but not in another of the same species. This is the case of *Solanum nigricans* (this study) and *Dryopteris wallichiana*, which could not be amplified in this study but were successfully amplified in a previous study using samples from different plants (*Trujillo-Argueta et al., 2021*). Another possibility is that the pair of *rbcL*a universal premiers used may not work for certain species. Our *rbcL*a amplification success (69%) could be increased using the alternative set of universal primers proposed by CCDB for gene barcode *rbcL*.

Our sequencing success (91.30%) was high and similar to those reported in other works. In a study of root diversity patterns using plastid gene *rbcL*a, *Kesanakurti et al. (2011)* registered 96% amplification success with 85% sequencing success. In another study that identified Sicily's most threatened plant taxa, the amplification and sequencing successes were 96% and 95%, respectively (*Giovino et al., 2016*). In a study of the temperate flora of Canada, the use of *rbcL*a gave a 91.4% sequencing success (*Burgess et al., 2011*).

Our BLAST results were higher for genus discrimination (80.77%) than the values obtained for species differentiation (15.38%). Results from other regions and species are variable. For example, in wild, arid plants, discrimination at genus and species levels were lower than ours: 50% and 8%, respectively (*Bafeel et al., 2012*), but higher in a comprehensive study of the local flora of Canada (91% and 44%, *Braukmann et al., 2017*). In a study of threatened species of Sicily, the discrimination at the genus level was lower (52%) but higher at the species level (48%) than our results (*Giovino et al., 2016*). The peculiarities of the biology of the studied species may also account for the observed discrimination variability. Part of our low percent species discrimination results using BLASTn can be explained by low marker resolution, as was noticed in those species that matched their *rbcL*a sequence with more than 30 different species in the GenBank database (*Alnus acuminate, Solanum hispidum, Quercus laurina, Quercus callophylla, Pinus montezumae, Osmanthus americanus* and *Physalis phyladelphica*). Another explanation is misidentified voucher specimens in public DNA databases, an issue that several authors have acknowledged (*e.g.*, *Burgess et al., 2011*; *Abdullah, 2017*). Since it is customarily to describe species based on morphological characteristics, it is possible that hybridization and polyploidy, which are common in plants, may contribute to decreasing barcoding

species discrimination (*Fazekas et al., 2008*; *Hollingsworth, 2011*). Since more than half of the species in this study (52.72%) lacked comparative data in the GenBank database, it is necessary to increase the DNA barcode database, particularly for tropical wild plant species. Indeed, we contributed to new 63 *rbcL*a sequences to BOLD, its metadata, and the GenBank database. Although 42 of our species already had a *rbcL*a sequence on the GenBank database, new records on these species might help discover new haplotypes or geographical variants (*Hajibabaei et al., 2007*). Even if *rbcL*a does not have high species discrimination, it does for genus discrimination, which for some ecological studies might be enough (*e.g.*, *Kesanakurti et al., 2011*).

Our distribution of intra- and interspecific genetic divergence (Fig. 2) agrees with the premise that a DNA barcode must exhibit high interspecific but low intraspecific divergence (*Lahaye et al., 2008*). The percent interspecific divergence of this study (0.65) is similar to those reported in other hotspot diversity areas such as the Mediterranean Basin (0.89) (*Giovino et al., 2016*) and Southern Africa (0.82) (*Lahaye et al., 2008*). The lack of genetic divergence observed in the three genera of trees: *Quercus* (*Q. martinezii, Q. laurina,* and *Q. callophylla*); *Oreopanax* (*O. sanderianus* and *O. xalapensis*), and *Daphnopsis* (*D. selerorum* and *D. tuerckheimiana*) concurs with *Smith & Donoghue (2008)*. These authors found that the rates of molecular evolution are low in woody plants with long generation times compared to herbs. In the case of oaks (*Quercus*), several attempts have been made to identify species in Italy, using different plastid barcodes without success since hybridization and polyploidy are expected to be high in this group (*Piredda et al., 2011*). Null genetic divergence obtained in *Oreopanax* and *Daphnopsis* (Table 3) is of concern since *Oreopanax sanderianus* and *Daphnopsis tuerckheimiana* are on the red list of IUCN. The highest values of genetic distance found in the Ericaceae (5.57%), Euphorbiaceae (4.59%), and Asteraceae (3.3%) families that hold many herbs and shrubs species agree with the assumption that the *rbcL*a barcode has a better species differentiation for non-tree species. Moreover, a study conducted in a subalpine forest in Southwest China found a better DNA barcode resolution for herbs than for tree species (*Tan et al., 2018*). However, more studies are needed to confirm this trend in other species and localities.

The phylogenetic arrangements found in our study using barcode *rbcL*a concur with the recent Angiosperm Phylogeny Group classification (APG IV) of flowering plants (*The Catalogue of Life Partnership, 2017*). The percent monophyletic species resolution obtained in this study using NJ (100%), ML (100%), and BI (85.45%) phylogenetic trees, was higher compared to 17% of species resolution found in arid wild plants using ML trees (*Bafeel et al., 2012*), barcoding the biodiversity of Kuwait (58%) using NJ trees (*Abdullah, 2017*) and the 71.8% registered in two biodiversity hotspots of Mesoamerica and Southern Africa, using ML and BI trees (*Lahaye et al., 2008*). Our ML phylogenetic tree showed the most robust phylogeny (87.27%), *Ocotea helicterifolia, Quercus callophylla, Quercus laurina, Iresine difussa, Berberis lanceolata, Moussonia deppeana*, and *Osmanthus americanus*, could not be resolved as monophyletic species with a clade bootstrap support value >70%.

Most of these species are trees in agreement with the assumption that rates of molecular evolution are low in woody plants compared to herbs (*Smith & Donoghue, 2008*). For those

species that could not be differentiated with the ML tree, we suggest the addition of a second barcode.

Species discrimination can be improved by using tree-based phylogenetic methods rather than BLAST analysis and genetic distance approaches. For instance, using NJ and ML phylogenetic trees, it was possible to differentiate *Quercus martinezii* from *Q. laurina* and *Q. callophylla* (Fig. S1, Fig. 3) despite the null genetic divergence observed in *Quercus* using BLAST. Based on an updated infrageneric classification of the oaks (*Denk et al., 2017*), *Q. martinezii* belongs to the white oaks (subsection *Quercus*), while *Q. callophylla* and *Q. laurina* belong to the red oaks (subsection *Lobatae*). In the Solanaceae family, three out of the five studied species (*Physalis philadelphica*, *Solanum hispidum*, and *Solandra maxima*) share high similitude with at least 30 species using the best BLAST match results. Furthermore, using our best BI tree, we observed a polytomy in the *Solanaceae* clade (Fig. S2), and a low discrimination value in the NJ tree. However, these species could be resolved with our ML phylogenetic tree. Taxonomic species are usually described based on morphological characteristics that can easily be altered by local adaptation, phenotypic plasticity, or neutral morphological polymorphism, which may cause a single variable species to be classified as many species (*e.g.*, *Gemeinholzer & Bachmann, 2005*). On the other hand, very recent divergence and little differentiation might contribute to the inability of barcoding to separate species in some cases (*Birch et al., 2017*).

## CONCLUSIONS

DNA barcoding using *rbcL*a can be a promising identification tool primarily at the family and genus level for vascular plant species of the neotropical montane cloud forest. We identify three major problems with the use of this technique. First, the lack of a universal amplification capability is probably associated with DNA degradation in some cases, but without ruling out other factors requiring further study. Second, the inability to detect certain morphological species is probably not related to *rbcL*a itself but to biological (*e.g.*, polyploidy and hybridization) and technical (misidentifications or taxonomic misclassifications) problems. Third, the few available registers in the BOLD and GenBank databases (more than half of our species, 52.72%, did not have previous *rbcL*a sequence records). Indeed, we contributed new 13 species to the GenBank Taxonomy Database and 63 new sequences for *rbcL*a in BOLD and GenBank. We found preliminary evidence suggesting that the ability of the marker to discriminate species is not randomly distributed among taxa. Herb and shrub species in the Asteraceae, Ericaceae and Euphorbiaceae families showed the highest genetic distance using *rbcL*a, which can be helpful to distinguish congeneric species. Contrastingly, we detected nil genetic divergence among congeneric species in long-lived tree genera, *Quercus*, *Oreopanax*, and *Daphnopsis*. Nonetheless, the accuracy for discriminating species can be substantially improved using tree-based analysis. While BLAST and genetic distance approaches could not differentiate *Quercus* species, NJ and ML could successfully separate white oaks (*Quercus martinezii*) from red oaks (*Q. callophylla* and *Q. laurina*). Also, most species in the Solanaceae family that showed unsuccessful BLAST results and low genetic distance could be discriminated

against with ML phylogenetic tree. The ML phylogenetic tree showed the most robust phylogeny (87.27%) of all our 55 studied species of the tropical montane cloud forest of San Miguel Cuevas in Oaxaca state, Mexico. The establishment of this local barcode database will be valuable for a broad range of potential ecological, conservational, and phylogenetic applications.

## ACKNOWLEDGEMENTS

Raúl Rivera García and Heladio Luna, assisted in field work.

### Funding

Funding was provided by Instituto Politécnico Nacional (Grant Nos. 20171792, 20180889, 20195601, 20201673 and 20211341). The funders had no role in study design, data collection and analysis, decision to publish, or preparation of the manuscript.

### Grant Disclosures

The following grant information was disclosed by the authors:
Instituto Politécnico Nacional: 20171792, 20180889, 20195601, 20201673 and 20211341.

### Competing Interests

The authors declare that they have no competing interests.

### Author Contributions

- Sonia Trujillo-Argueta conceived and designed the experiments, performed the experiments, analyzed the data, prepared figures and/or tables, authored or reviewed drafts of the article, and approved the final draft.
- Rafael F. del Castillo conceived and designed the experiments, analyzed the data, authored or reviewed drafts of the article, and approved the final draft.
- Abril Velasco-Murguía analyzed the data, authored or reviewed drafts of the article, and approved the final draft.

### Field Study Permissions

The following information was supplied relating to field study approvals (*i.e.*, approving body and any reference numbers):

The community local authorities of San Miguel Cuevas Municipality granted verbal permission to work in their lands. Two commissioners of communal property, Mr. Pedro Gil (2017-July 2018) and Mr. Damián Domínguez (July 2018 to June 2019) were directly responsible for such permissions.

### DNA Deposition

The following information was supplied regarding the deposition of DNA sequences:
The sequences are available at GenBank: MZ771330, ON002485–ON002546.

## Data Availability

The metadata and DNA sequences are available at BOLD registered under project name Diversity of a humid temperate forest in Oaxaca, Mexico (project code DVHTF; http://www.boldsystems.org).

The DNA sequences are available at GenBank and the raw data are available in the Supplementary File.

## Supplemental Information

Supplemental information for this article can be found online at http://dx.doi.org/10.7717/peerj.13771#supplemental-information.

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
