# Peer review of "Testing the effectiveness of rbcLa DNA-barcoding for species discrimination in tropical montane cloud forest vascular plants (Oaxaca, Mexico) using BLAST, genetic distance, and tree-based methods"

_PeerJ, doi:10.7717/peerj.13771_

## Round 0.1 · original submission · Minor Revisions

Please find below the comments by three reviewers. I coincide especially with the comments of Reviewer 3 in that interpretation of results and discussion need major revision. Also, add justification for the use of one marker. Reviewer 3 included an excellent review made directly in the manuscript file, please when writing your response letter indicate how you dealt with every issue.

Reviewer 1 ·

Basic reporting

1. Citation should be arranged according to chronological time (line 59-60). Please check it in other places in the manuscript.

2. What is rbcLa? I suggest the authors provide figure the structure of rbcLa, so that reader could understand comprehensively on the marker used.

3. Grammar issue line 88. Please check it in other places.

Experimental design

1. What do you mean with “plastid marker” in line 63? Plastid is one of genome in the plant not marker.

2. What do you mean with one sentence in line 71-73. Ecosystem vs temperate region?

3. Confusing statements in last paragraph of Introduction. Do you thing that there is connection between locality/region/ecosystem and rbcLa marker, so that you selected this marker? I dont think so.

Validity of the findings

1. Discussion part need to be elaborated more.

Additional comments

1. In general, this manuscript is well prepared and highly contributes to the science of DNA barcoding for plant
2. Citation should be arranged according to chronological time (line 59-60). Please check it in other places in the manuscript.
3. What do the authors mean with “plastid marker” in line 63? Plastid is one of genome in the plant not marker.
4. What is rbcLa? I suggest the authors provide figure the structure of rbcLa, so that reader could understand comprehensively on the marker used.
5. What do the authors mean with one sentence in line 71-73. Ecosystem vs temperate region?
6. Confusing statements in last paragraph of Introduction. Do the authors thing that there is connection between locality/region/ecosystem and rbcLa marker, so that you selected this marker? I dont think so.
7. Grammar issue line 88. Please check it in other places.
8. Discussion part need to be elaborated more.

Annotated reviews are not available for download in order to protect the identity of reviewers who chose to remain anonymous.

·

Basic reporting

the manuscript is very informative but needs an addition for some points as suggested in the notes.

Experimental design

the experiments need an addition in methods as suggested in the text, such as the species selection, what kind of leaf taken for analysis, etc.

Validity of the findings

no comment

Additional comments

the manuscript is written in a good way with a brief explanation. but, there are some pieces of information that need to be added to make the manuscript goes the way it should. some common mistakes found in the manuscripts are the words for rbcL, matK which were mistakenly typed and the references that are not on the list but they are listed.

Reviewer 3 ·

Basic reporting

The manuscript is easy to read with few areas where the written content is hard to follow.

There are a number of internet text space characters throughout the document that have not been removed or corrected.

One strength would have been to look at both rbcL and MatK for these samples, but I understand that both time and funding to complete this may not have been possible. And contributing a single marker to our greater understanding of DNA barcoding for this geographic region and the taxa collected is commendable.

For specific comments please see my attached pdf document.

Experimental design

The experimental design was fairly straightforward with the collection of taxa at a particular site and then the morphological identification and DNA barcoding of collected specimens at that site.

While the design seems robust enough, my concerns are with the lack of reporting on how the study design was constructed and conducted. Why was the specific site selected? What is the geographic range of these collections? What was the expected coverage of the taxa present at this site represented in the dataset for this manuscript?

Validity of the findings

The specifics of the application of the barcoding and barcode gap analyses need to be revisited. Also, the specifics of the tree-based analyses and the application of the analyses at differing taxonomic levels are not consistent with the data presented.

Additional comments

The samples represented in this work seem to be sufficient for a contribution to this area of science. While the specific testing of the rbcL molecular marker as a single marker for plant identification is not novel, the combination of the sampled species and geographic region does provide a compelling story and as such on a whole this work seems novel enough for publication.

I do think that this study is interesting and could be contributory to the area of DNA barcoding of plants in the geographic region studied. However, I think that the reporting of these data and the interpretation and discussion of the results within this study need to be revisited. I recommend that this work be revised, rewritten or corrected where appropriate and resubmitted. Please see my attached pdf with additional comments.

Annotated reviews are not available for download in order to protect the identity of reviewers who chose to remain anonymous.

---

## Round 0.2 · accepted · Accept

Thank you for considering the last issues on your paper. Please make sure that the sequences submitted to GenBank and BOLD are public.